# Fungal–bacterial interaction selects for quorum sensing mutants with increased production of natural antifungal compounds

Andrea G. Albarracín Orio [1,2,3,4✉], Daniel Petras [4,5,6,7], Romina A. Tobares [2,3,7], Alexander A. Aksenov[4,6], Mingxun Wang[4,6], Florencia Juncosa [1], Pamela Sayago [1], Alejandro J. Moyano [2,3], Pieter C. Dorrestein[4,6✉] & Andrea M. Smania [2,3✉]

Soil microorganisms coexist and interact showing antagonistic or mutualistic behaviors. Here, we show that an environmental strain of *Bacillus subtilis* undergoes heritable phenotypic variation upon interaction with the soil fungal pathogen *Setophoma terrestris* (ST). Metabolomics analysis revealed differential profiles in *B. subtilis* before (pre-ST) and after (post-ST) interacting with the fungus, which paradoxically involved the absence of lipopeptides surfactin and plipastatin and yet acquisition of antifungal activity in post-ST variants. The profile of volatile compounds showed that 2-heptanone and 2-octanone were the most discriminating metabolites present at higher concentrations in post-ST during the interaction process. Both ketones showed strong antifungal activity, which was lost with the addition of exogenous surfactin. Whole-genome analyses indicate that mutations in ComQPXA quorum-sensing system, constituted the genetic bases of post-ST conversion, which rewired *B. subtilis* metabolism towards the depletion of surfactins and the production of antifungal compounds during its antagonistic interaction with *S. terrestris*.

[1] IRNASUS, Universidad Católica de Córdoba, CONICET, Facultad de Ciencias Agropecuarias, Córdoba, Argentina. [2] Universidad Nacional de Córdoba. Facultad de Ciencias Químicas. Departamento de Química Biológica Ranwel Caputto, Córdoba, Argentina. [3] CONICET. Universidad Nacional de Córdoba. Centro de Investigaciones en Química Biológica de Córdoba (CIQUIBIC), Córdoba, Argentina. [4] Collaborative Mass Spectrometry Innovation Center, University of California San Diego, La Jolla, CA 92093, USA. [5] Scripps Institution of Oceanography, University of California San Diego, La Jolla, CA 92093, USA. [6] Skaggs School of Pharmacy and Pharmaceutical Sciences, University of California, San Diego, La Jolla, CA, USA. [7] These authors contributed equally: Daniel Petras, Romina A. Tobares. ✉email: andrea.albarracin@gmail.com; pdorrestein@ucsd.edu; asmania@unc.edu.ar

The great diversity of soil microbes leads to extensive interspecies interactions[1]. Interactive microbial communities are one of the main factors that influence soil fertility, so playing a key role not only in the ecosystem's health, but also in agricultural production, which pursues to meet the needs of a growing world population. Thus, in the context of soil management, a better understanding of microbial interactions is essential to fully exploit the benefits provided by soil organisms.

Antagonistic and mutualistic behaviors, mediated by the exchange of small diffusible secondary metabolites, facilitate microbial adaptation to the complex communal lifestyles. Among them, fungi and bacteria are found living together in a wide variety of soil environments frequently involved in complex interactions shaped by several molecular determinants, such as motility, quorum sensing (QS), bacterial secretion system, and secondary metabolites[2,3].

*Bacillus subtilis* is a remarkably diverse Gram-positive bacterial species that is commonly found in the upper layers of the soil, as well as in plant rhizosphere[4]. The rich amount of genetic pathways for the utilization of plant-derived molecules supports the fact that this species develops closely associated with plants[5]. Moreover, the ability of *B. subtilis* to sense small molecules produced by a wide range of soil microorganisms indicates the presence of broad response mechanisms to neighboring species. In this sense, *B. subtilis* became an interesting model to develop strategies for the biological control of a wide range of organisms, such as bacteria[6], fungi[7,8], nematodes[9], or even insects[10,11]. This capacity stems from the numerous metabolites secreted by *B. subtilis*, of which many play dual roles as antimicrobial compounds and signaling molecules, participating in processes, such as regulation of development, biofilm formation, and inhibition of virulence factors released by competitors[12–14]. In fact, whole-genome sequencing revealed that *B. subtilis* devotes almost 4% of its genome to making secondary metabolites[5]. In *B. subtillis*, like in most bacterial species, the regulation of many of these functions and others, such as competence for DNA uptake, sporulation, and biofilm development, are regulated by the QS system[15]. Interestingly, bacterial QS systems are not restricted to communication within their own species but are capable of receiving signals from and/or sending them to unrelated species, which might end up into behavioral modifications of one or more interacting partners[16].

We previously isolated from the rhizosphere of onion plants a strain of *B. subtilis* (referred to as Bs ALBA01) that strongly inhibited in vitro growth of the soil fungus *Setophoma terrestris*, a major plant pathogen that affects several economically important vegetable crops[17]. Particularly in onion, *S. terrestris* causes the so-called pink root, the most severe onion disease in soils from subtropical and tropical regions[18]. Interestingly, this response seems to be specific against *S. terrestris* since cocultures with other pathogens, such as *Fusarium* species did not show growth inhibition or behavioral changes[17]. Importantly, antifungal activity was observed for cell-free culture supernatants of Bs ALBA01 previously grown in the presence of *S. terrestris* (post-ST variants), but not of Bs ALBA01 grown in the absence of the fungus (pre-ST variants). Moreover, the antagonistic compounds secreted by post-ST variants caused several alterations in the morphology and structure of hyphae of *S. terrestris* that could result in the cell growth and reproduction arrest[17]. However, the mechanistic bases of this phenomenon and the nature of the antifungal compounds remained unexplained. In the present study, a combined approach using phenotypic characterization, untargeted liquid chromatography tandem mass spectrometry (LC-MS/MS)-based metabolomics, gas chromatography–mass spectrometry (GC–MS)-based metabolomics, and comparative genomics demonstrated that *B. subtilis* undergoes mutation-based

phenotypic variation, which is triggered only upon interaction with *S. terrestris*. These changes involve the production of 2-ketones as the main antifungal compounds and increased biofilm formation. Paradoxically, antifungal activity also required the loss of surfactin and plipastatin biosynthesis, two cyclic lipopetides widely recognized as the main antimicrobials that can be secreted in biologically relevant amounts[19,20]. Mutations in the ComQXPA QS system were involved in the stable phenotypic and metabolomics profile changes necessary to thrive upon the antagonistic interaction with the fungus.

Our results advance the understanding of the biological interaction strategies of *B. subtilis*, which will help to elucidate the regulatory systems involved in the expression of metabolites for chemical communication and biocontrol activity.

## Results and discussion

### *B. subtilis* stably changes phenotype against *S. terrestris*. We have previously observed that *B. subtilis* (referred to as Bs ALBA01) isolated from the rhizosphere of onion plants inhibits the growth of the soil fungal pathogen *S. terrestris*[17]. After 15 days of interaction with the soil fungal pathogen *S. terrestris* in coculture assays, *B. subtilis* acquired strong antifungal activity. Such activity was mediated by secreted factors, since bacterial cell-free supernatants displayed the fungal growth inhibition[17] (Fig. 1a, e). These Bs ALBA01 variants were termed post-ST, to distinguish them from pre-ST variants grown without any contact with the fungus and whose cell-free supernatants displayed no antifungal activity[17] (Fig. 1a, e). Comparisons between pre-ST and post-ST variants revealed clear phenotypic differences, with post-ST variants showing greater degrees of roughness and wrinkled colony phenotype, thicker and more structured pellicles associated with robust biofilm formation and, lack of swarming motility (Fig. 1b–e).

Importantly, all post-ST phenotypic traits were maintained when variants were reinoculated onto the fresh growth medium (passaging), indicating that *B. subtilis* experienced genetically stable phenotypic variation upon coculture with *S. terrestris*.

### Post-ST variants differ in metabolome and reduced surfactin. We next characterized the metabolites profile changes suffered by *B. subtilis* due to conversion to post-ST, which might be responsible for the antagonistic effects observed on *S. terrestris*. Thus, by using nontargeted metabolomics profiling based on $^1$H-NMR spectroscopy, we analyzed cell-free supernatants from pre- and post-ST. Pre-ST samples clustered together and were clearly separated from post-ST samples (Supplementary Fig. 1a, c, d). S-line plots in orthogonal partial least squares discriminative analysis (OPLS-DA) models showed clearly distinct signals, representing discriminant metabolites between the pre- and post-ST sample groups (Supplementary Fig. 1b).

Furthermore, we also compared chemotypes of whole cells in search of significant metabolomics differences between the two variants. Nontargeted high-performance liquid chromatography tandem mass spectrometry (HPLC–MS/MS)-based metabolomics analysis revealed separation of global metabolome between the two variants in principal coordinates analysis (PCoA; Fig. 2a). Surprisingly, the two well-known *Bacillus* lipopeptides, surfactin and plipastatin, were severely reduced in the post-ST variant (Supplementary Fig. 2a, b). This finding was intriguing because surfactin and plipastatin are two of the most important compounds employed by *B. subtillis* to thrive in microbial antagonistic interactions[19,20]. In fact, random forest analysis showed that they were the most important variables determining differences between pre- and post-ST (Supplementary Fig. 2c). Post-ST showed notable reduction in intensity of surfactin ions

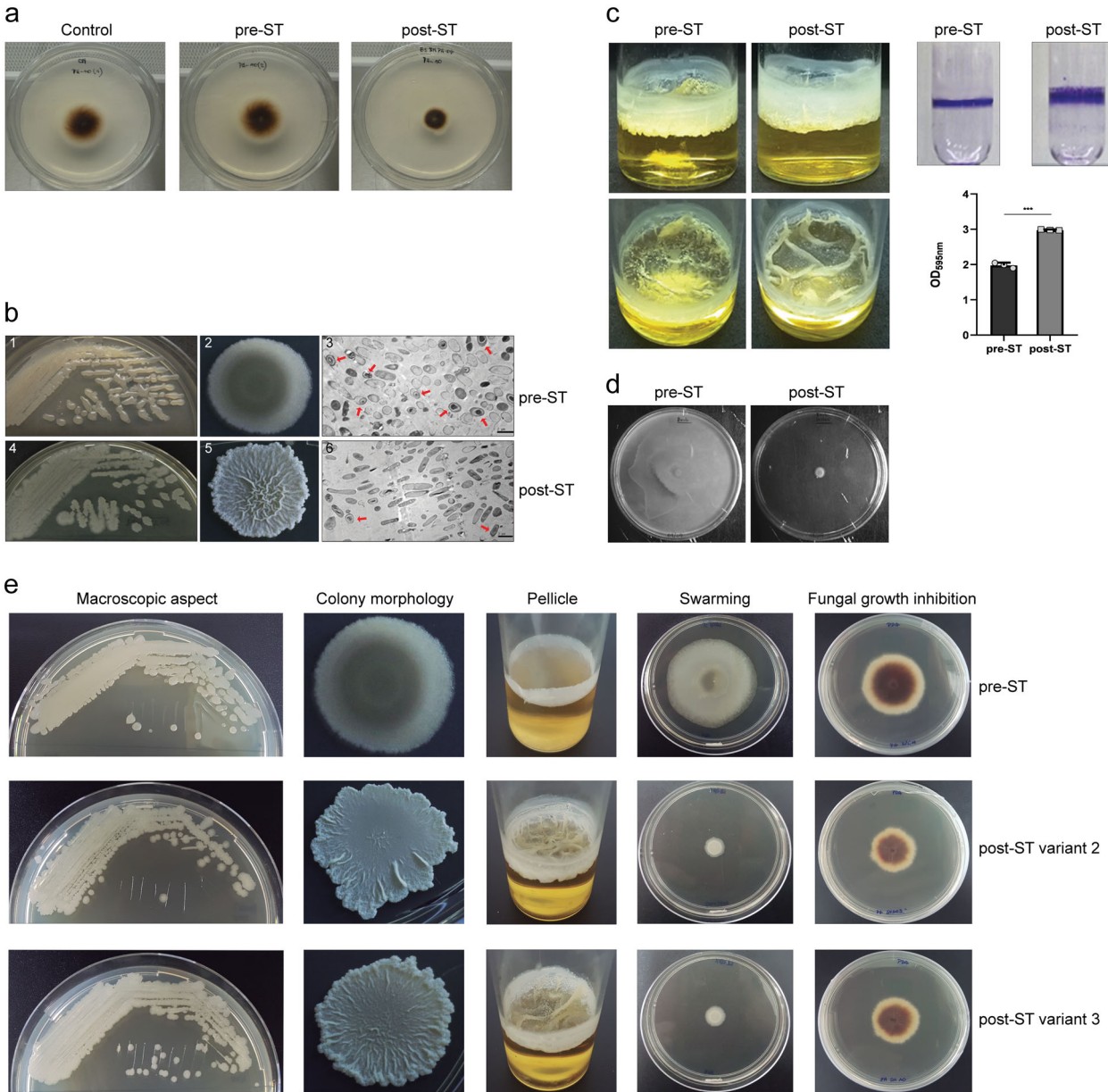

**Fig. 1 Coculture of *B. subtilis* with *S. terrestris* induces stable phenotypic changes. a** Growth inhibition of *S. terrestris* at day 4 after inoculation. Strong growth inhibition was observed in dishes containing only cell-free supernatants of post-ST. **b** Phenotypic changes of Bs ALBA01 resulting from interaction with fungus on LB agar culture. Colonies of post-ST were rougher and more wrinkled than those of pre-ST. Agar plates (panels 1 and 4) and individual colonies (panels 2 and 5) of pre- and post-ST, respectively. Electron microscopy revealed increased numbers of elongated cells and reduced numbers of sporulated cells in post-ST (panel 6) relative to pre-ST (panel 3). Scale bar: 2 μm. Arrows: sporulated cells. **c** Post-ST variants in liquid medium developed thick, wrinkled pellicles characteristic of strong biofilm formation. Quantification of biofilm formation on borosilicate surfaces by crystal violet staining revealed greater biofilm formation by post-ST than by pre-ST. Data shown are mean values of measurements of optical density at 595 nm of crystal violet suspensions from three independent replicate experiments. ***Significant difference between values ($p < 0.0001$, Tukey's multiple comparison test). **d** Swarming motility assessed on 0.7% agar LB plates. Pre-ST displayed full motility (i.e., covered the entire plate), whereas post-ST showed no swarming motility (i.e., did not extend beyond the inoculation area). **e** All *B. subtilis* variants obtained following coculture with *S. terrestris* acquired similar stable phenotypes. The panel shows phenotypes of other two post-ST, post-ST variants 2 and 3, and variants whose genomes were sequenced. Similarly to observations of post-ST, post-ST2 and 3 showed rough, wrinkled colony appearance, and robust pellicle formation relative to pre-ST. Swarming motility was absent in post-ST2 and 3. Dishes containing only cell-free supernatants of post-ST2 and 3 showed strong growth inhibition of *S. terrestris* (7 days after inoculation, at 30 °C).

and of peaks with masses indicative of plipastatin, identified through spectrum library matching and feature-based molecular networking[3,21,22]. Importantly, post-ST clustered together with a *B. subtilis* NCBI 3610 surfactin-defective mutant[23], confirming the role of surfactin in separation between the groups (Fig. 2a).

Molecular networks obtained by analysis of chemical profiles of the two variants indicate reduction in post-ST of not only surfactin and plipastatin derivatives, but also related unknown derivatives (Supplementary Fig. 3). We further complemented this analysis with HPLC–MS/MS-based metabolomics analysis of

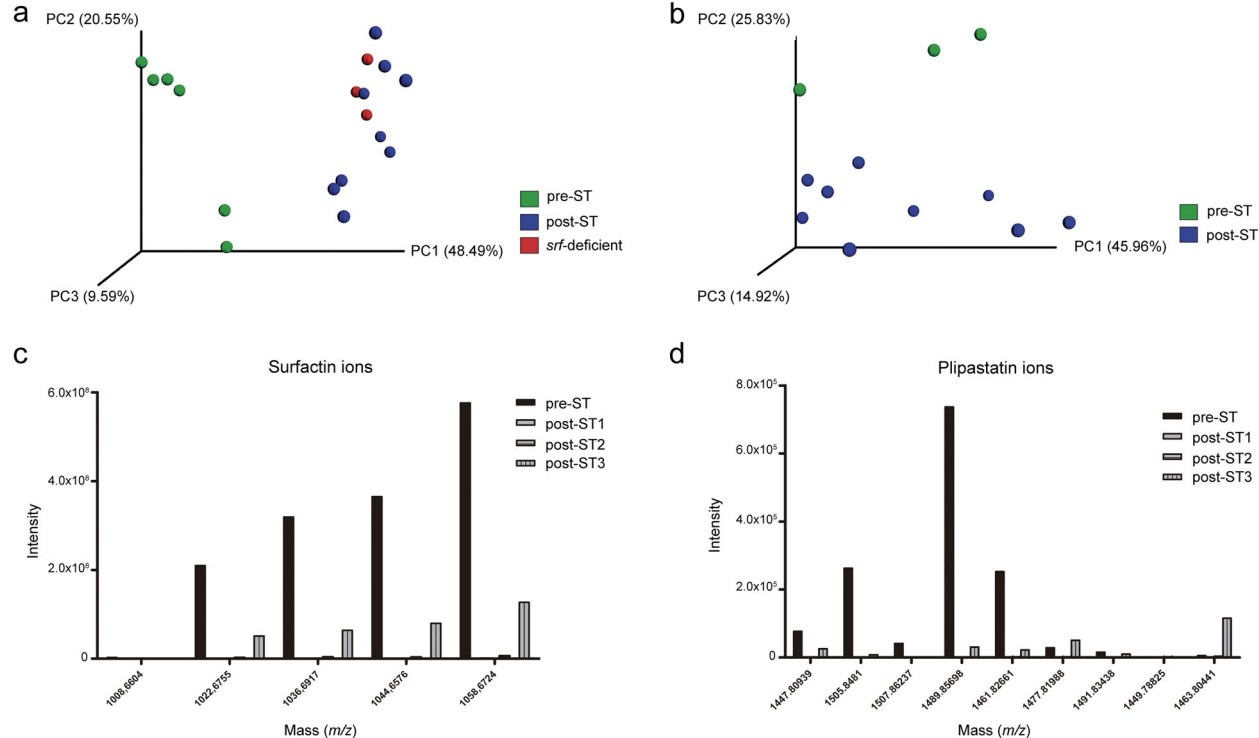

**Fig. 2 LC-MS/MS-based metabolomics analysis of whole cells and cell-free supernatants showing chemical signatures that distinguish pre- from post-ST variants. a** PCoA plots show strong separation between whole-cell sample analysis of pre- and post-ST. **b** PCoA plots show clear distinctions according to the origin of cell-free supernatants, with separate clustering of pre- vs. post-ST samples. Post-ST variants clustered together and separately from the pre-ST ancestral variant. The group of post-ST samples in PCoA plots included samples of three independently obtained post-ST variants (post-ST1, post-ST2, and post-ST3) after coculture with *S. terrestris* in vitro (see "Methods" for details). Metabolomics experiments were carried out in triplicates for each post-ST variant. **c** Intensities of surfactin and **d** plipastatin ions were strongly reduced in cell-free supernatants of post-ST.

cell-free supernatants of the two variants. Once again, PCoA plots evidenced a discrimination according to the origin of cell-free supernatants, displaying separate clustering of pre-ST vs. post-ST samples (Fig. 2b). Consistently with results of whole-cell analysis, supernatants of post-ST did not show surfactin (Fig. 2c) or plipastatin (Fig. 2d) ions levels comparable to those of pre-ST. Accordingly, post-ST supernatants showed no hemolytic activity (Supplementary Fig. 2d).

Since the absence of surfactin in post-ST was unexpected, we decided to evaluate the role of surfactin deficiency on *B. subtilis* in this interaction process. To examine the possibility that key genes involved in surfactin production are related to acquisition of antifungal activity by post-ST variants, we generated a surfactin-defective Bs ALBA01 mutant (hereafter Bs *srfAA*) by disrupting the surfactin synthase encoding gene *srfAA*, essential component of the machinery for non-ribosomal synthesis of surfactin. Anti-*S. terrestris* activity of this mutant was screened upon 15-day of coculture by obtaining and testing samples before and after coculture. Interestingly, the absence of surfactin resulted in the acquisition of equivalent mycelial growth inhibitory activity to that observed in post-ST. In fact, pre- and post-ST supernatants of surfactin-defective Bs *srfAA* showed no notable differences in antifungal activity (Supplementary Fig. 4). Thus, inability to produce/release surfactin resulted in enhanced anti-*S. terrestris* activity, as well as complete suppression of ST-driven transformation to post-ST phenotype. On the other hand, unlike the post-ST variants and in agreement with previous reports[6], the Bs *srfAA* strain lost the capacity to form biofilms.

**2-Ketones mediate growth inhibition of *S. terrestris*.** Given that metabolomics profiles from HPLC–MS/MS showed clear

differences, it was intriguing that the main difference in the metabolome of post-ST variants was the suppression of lipo-peptides surfactin and plipistatin, whereas no candidate anti-fungal metabolites showed significantly higher abundances. However, our HPLC–MS/MS analysis was biased toward a limited area of the chemical space and was unable to detect small primary metabolites and volatile compounds. In order to expand our search to small molecules, we analyzed the volatile compounds profile of the variants and their interaction with *S. terrestris*. Thus, we adapted the growth of cocultures to glass vials suitable for headspace GC–MS analysis using solid-phase microextraction (SPME). From PLS-DA, we found that 2-heptanone was the most discriminating compound produced in greater amount post-mutation only by *B. subtilis*, and not by the fungus. Interestingly, by performing a GC data molecular network analysis[24] within the Global Natural Products Social (GNPS) platform[22], we could discriminate a cluster composed of a family of ketones, remarkably, all 2-ketones (Fig. 3a). Several of these ketones were also among the top discriminant features between pre-ST and post-ST variants and *S. terrestris* (Fig. 3b). Interestingly, individually cultured pre-ST variants showed high levels of 2-ketones (Fig. 3b). However, coculture with *S. terrestris* was able to induce a steep drop in 2-ketones levels in pre-ST, but remarkably post-ST variant retained the ability to produce them. Thus, we wondered whether 2-heptanone and a representative compound of the ketone cluster, 2-octanone, were sufficient to exert antifungal activity on *S. terrestris*. To the best of our knowledge, there is no specific genic pathway described in *B. subtilis* responsible for the synthesis of 2-ketones and thus, no 2-heptanone- nor 2-octanone-deficient mutants of *B. subtilis* can be achieved. Therefore, we decided to test the antifungal effect of

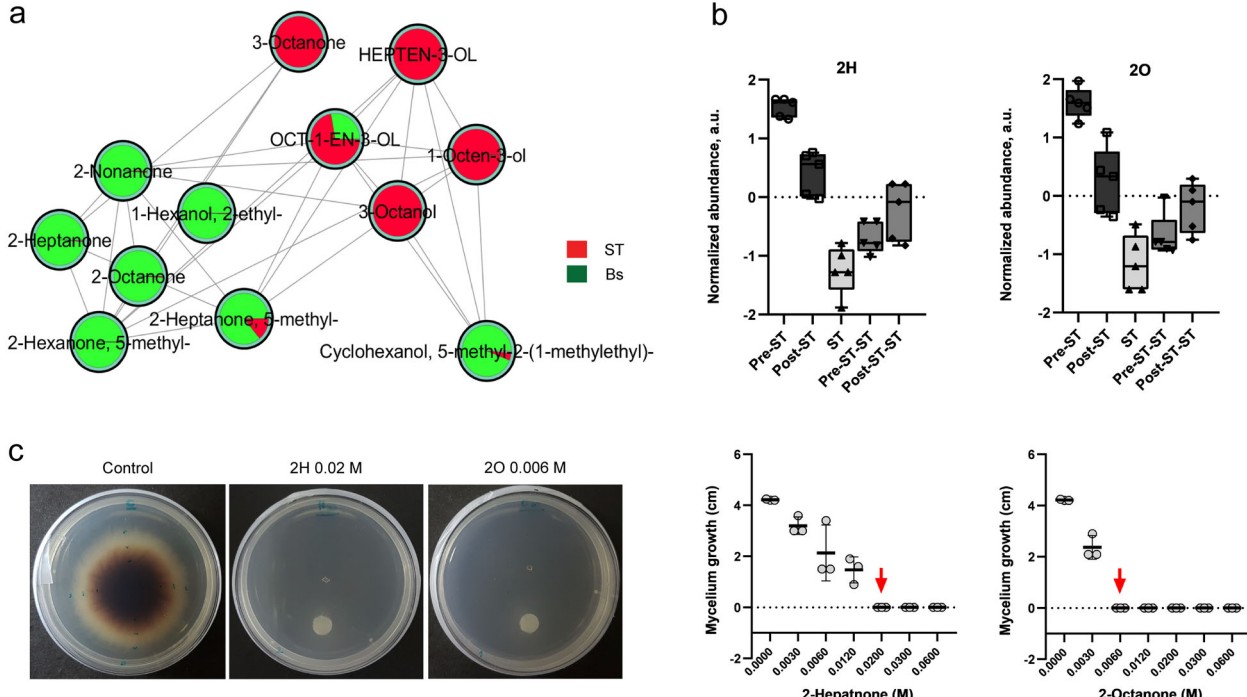

**Fig. 3 Volatile compounds change due to interaction between *B. subtilis* and *S. terrestris*: 2-ketones induce fungal growth inhibition. a** GC molecular network analysis revealed that Bs ALBA01 produces and releases a family of 2-ketones compounds, which arise as the most important volatile metabolites produced only by the bacterium. **b** Interaction with the fungus (ST) generates a steep drop of 2-heptanone (2H) and 2-octanone (2 O) levels in pre-ST, but not in post-ST. Both 2H and 2O are the most discriminating features between pre- and post-ST during coculture with ST. Boxplots show compound abundances, normalized by quantile normalization and auto-scaled (zero-centered and divided by SD). **c** Growth inhibition of *S. terrestris* at day 7 after inoculation of 0.02 M 2H and 0.006 M 2O on a filter paper disc placed on PDA dishes. Graphs showing the gradual reduction in mycelium growth with increasing concentrations of 2-ketones. Data shown are mean values of mycelial growth from three independent replicate experiments; red arrow indicates lethal concentration of 2H and 2O at day 7 after inoculation.

either 2-ketones on *S. terrestris*. The activity of both compounds was tested in bioassays by culturing the fungus on potato dextrose agar (PDA) dishes containing a filter paper disc, where different concentrations of each purified compound were loaded. The growth of *S. terrestris* was gradually reduced as the concentration of 2-heptanone and 2-octanone increased, and the fungal inhibition was maximum at concentrations of 0.02 and 0.006 M, respectively (Fig. 3c). These results indicate that 2-heptanone and 2-octanone were involved in the antagonistic process against *S. terrestris*.

**Surfactin suppresses the antifungal activity of 2-ketones**. Considering our HPLC–MS/MS results, the antifungal behavior of the Bs *srfAA* mutant strain described above and the lethal effect of both volatiles 2-heptanone and 2-octanone on *S. terrestris*, we next explored whether the presence of surfactin was involved in the lack of antifungal activity characteristically observed in cell-free supernatants of pre-ST variants. If this were the case, then surfactin depletion would be a functional trait of post-ST variants necessary to achieve antifungal capacity. To evaluate this, we added exogenous purified surfactin to cell-free supernatant of post-ST and tested the anti-*Setophoma* activity of this combination. Exogenous surfactin partially suppressed the antifungal activity of post-ST supernatants (Fig. 4a). Furthermore, in order to distinguish if this suppressor effect was the outcome of surfactin directly interfering with the activity of antifungal compounds, we tested if surfactin was able to decrease the anti-*Setophoma* activities of 2-heptanone and 2-octanone. The exogenous addition of surfactin protected *S. terrestris* from the

antifungal activity of both, 2-heptanone and 2-octanone (Fig. 4b and Supplementary Fig. 5).

Taking into consideration that post-ST variants can maintain effective levels of 2-ketones when interacting with the fungus, we suggest that surfactin interferes with 2-ketone activity, probably by direct chemical reaction and/or by physical sequestration through micelle formation. Alternatively, some authors hypothesize that surfactin may have a stabilizing effect on the lipid bilayers in the fungal membrane generating some kind of resistance to anitifungal compounds[13]. Yet, the underlying mechanisms of surfactin interference with 2-ketones or other compounds remain unexplored and need to be further investigated.

In line with our results, previous reports proposed that surfactins could interfere with the activity of other lipopeptides with documented antifungal activity[13,25]. By studying the efficacy of *Bacillus amyloliquefaciens* strains at inhibiting *Rhizomucor variabilis*, a fungal pathogen of maize plants, Zihalirwa Kulimushi et al.[13], showed that fengycin and/or iturins, but not surfactin provided the main antifungal potential against *R. variabilis*. However, they revealed that the coproduction of surfactin together with fengycin decreased the global antifungal activity. Interestingly, the authors speculate that fengycins may be involved in permeabilization of spore/conidia and, therefore, inhibiting germination and/or causing hyphal cell perturbation. Opposing this, surfactin may have a stabilizing effect on certain lipid bilayers, thus limiting pore formation in fungal membranes. They also propose that under certain conditions and concentrations, surfactins and fengycins may co-aggregate, and form inactive complexes. Moreover, Kim et al.[25] reported that a

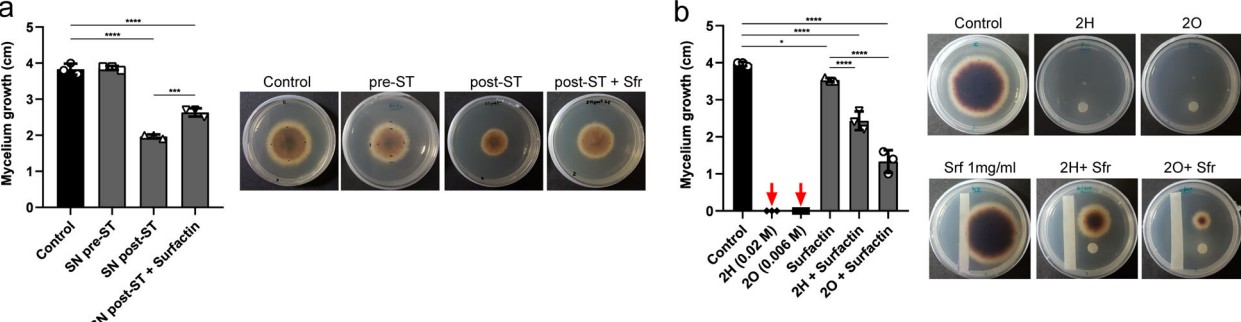

**Fig. 4 Surfactin interferes with the anti-*S. terrestris* activity of 2-ketones. a** Suppressor effect of exogenous surfactin (1 mg/ml) on the antifungal activity of cell-free supernatants of post-ST observed at day 5 after inoculation. **b** Suppressor effect of a filter paper strip imbibed with surfactin (1 mg/ml) on the antifungal activity of 2H (0.02 M) and of 2O (0.006 M), 7 days after inoculation. Data shown are mean values of mycelial growth from three independent replicate experiments; red arrow indicates lethal concentration of ketones at day 7. Statistically significant differences at $p < 0.0001$, $p < 0.001$, and $p < 0.05$ are identified by ***, **, and *, respectively (one-way ANOVA followed by Tukey's multiple comparison test).

surfactin-deficient mutant of *B. subtilis* subsp. *krictiensis* ATCC 55079 showed increased levels of lipopeptide iturin A and higher antifungal activity against the fungus *Fusarium oxysporum*, compared to its parental wild-type strain. The authors suggested that higher antifungal activity was achieved not only by the enhanced production of iturin A, but also by the absence of surfactin, which could be either blocking the iturin A production by sequestering substrates and/or by changing the activity of positive/negative regulators of cyclic lipopeptides.

**Post-ST variants show loss-of-function mutations in QS genes.** Finally, in order to elucidate the genetic bases of the phenotypic changes underwent by *B. subtillis* upon interaction with *S. terrestris*, we sequenced whole genomes of three post-ST variants (post-ST1, post-ST2, and post-ST3; Fig. 1e) derived independently from the Bs ALBA01 strain, which was considered as pre-ST ancestor and used as reference. Reads from these three variants were aligned against Bs ALBA01 genome[26] (NCBI Bioproject PRJNA316980) to assess genetic changes accumulated in post-ST during interaction with the fungus. We found one to three different mutations in coding sequence regions for each of the variants (Supplementary Table 1). Interestingly, each variant had at least one loss-of-function mutation in genes of the ComQXPA QS system. The post-ST1 and post-ST2 variants had a frameshift mutation in coding sequence of the *comA* gene, while post-ST3 variant harbored a 100 bp deletion in the kinase-encoding gene *comP*. Mutations in *comA* and *comP* were further confirmed by PCR amplification and sequencing.

Then, we investigated the extent to which *com* genes in Bs ALBA01 were convergently mutated upon interaction with *S. terrestris* by performing whole-genome sequencing of a pool of 15 post-ST variants obtained independently, following interaction with the fungus in coculture (Pool-seq; see "Methods"). To identify allelic variations in the pool, we mapped sequence reads to the *B. subtilis* ALBA01 reference genome, and then performed variant calling (see Supplementary Methods). Of note, six different loss-of-function mutations were identified in the *comQXPA* operon within the pool of post-ST variants, revealing marked convergence of mutation within these genes (Supplementary Table 1). These findings demonstrate the occurrence of mutation-based phenotypic changes in post-ST variants and support the notion that interaction of *B. subtilis* with *S. terrestris* induces metabolomics changes promoting bacterial antifungal activity, which involves the ComQXPA QS system. In addition, they suggest that single mutations in one of the ComQXPA genic components provided pleiotropic effects for the adaptation of *B. subtilis* to antagonistic interactions.

The ComQXPA system has been reported to be required for transcription of numerous genes involved in competence development, antibiotic production, exopolysaccharide production, degradative enzyme production and transport, and fatty acid metabolism[27,28]. In fact, over 10% of the *B. subtilis* genome is controlled by the ComQXPA QS system[28]. Mutations in ComA could possibly provoke alterations on the general metabolism promoting chemical conditions that favor the biosynthesis of 2-heptanone and 2-octanone, as those produced through the degradation of branched-chain amino acids, leucine, valine, and isoleucine, which could otherwise be used for surfactin or lipopetides synthesis. Alternatively, it has also been described that ComA controls the expression of FapR, a transcriptional regulator involved in fatty acid synthesis in *B. subtilis*[28]. In this sense, FapR negatively regulates the expression of at least ten genes (the *fap* regulon)[29]. Thus, mutations in *comA/comP* could determine an increased fatty acid biosynthesis by downregulating FapR. The release of the *fap* regulon in association to the chemical condition derived from the global change produced by the mutation may be favoring branched-chain fatty acid biosynthesis pathways and/or decarboxilation of intermediates ketoacids that promote the synthesis of 2-ketones. Accordingly, a recent report showed an enhancement of de novo fatty acid synthesis in *Bacillus nematocida* B16 during 2-heptanone production[30].

Notably, ComA activates transcription of the *srfA* operon that governs surfactin production[31,32]. The ComQXPA system has also been reported to play a role in transcription of regulator DegQ, which controls the production of degradative enzymes and is required for plipastatin production[33–35]. Thus, mutations in ComQXPA are in agreement with the loss of surfactin and plipastatin production observed in post-ST.

Until recently, it was widely accepted that production of surfactin was essential for biofilm formation. Thus, the enhanced biofilm formation displayed by post-ST variants resulted at first contradictory. However, it has been recently described that *B. subtilis* strains carrying mutations in genes of the ComQXPA QS system actually form pellicles with more matrix components than the wild-type strain[36]. In fact, ComQXPA mutants are able to develop earlier and thicker biofilms even though with decreased surfactin production. In agreement with these antecedents, we hypothesize that post-ST variants carrying loss-of-function mutations in ComQXPA system, promote the biofilm formation as an adaptive strategy to overcome interaction with the fungus. In addition, another study which described the diversity of ComQXPA among isolates from the tomato rhizoplane, also revealed a remarkable diversity in the surfactin production, which was not always positively correlated with the biofilm formation[37].

The authors observed that *srfA* gene inactivation in different natural isolates of *B. subtilis* showed different effects on biofilm biomass production, either displaying lower, equal, or higher biofilm formation than the wild-type strain. These observations indicate that the regulatory role of surfactin in biofilm formation may vary among different natural isolates of *B. subtilis*.

Among many other functions, it has been shown that inactivation of QS can lead to a delay in sporulation entry[36], which is in accordance with our observation of a reduced sporulation level observed in post-ST variants (Fig.1b). Interestingly, it has been discussed that ComQXPA system may act as a switch that contributes to the stochastic initiation of sporulation which, consequently, is able to achieve a bet-hedging behavior to limit the investment in population growth, and favor commitment to late growth adaptive processes[36].

## Concluding remarks

Here, we describe the strategies employed in the warfare undertaken by a soil strain of *B. subtilis* during the antagonistic interaction with the plant fungal pathogen *S. terrestris*. A bacterial heritable phenotypic variation and a metabolic shift occurred upon interaction with the fungus, which were mediated by mutations in the ComQXPA QS system. To our knowledge, this work uncovers the ComQXPA QS system as a novel pathway employed by *B. subtilis* to achieve stable metabolic rewiring, which enables the production of extracellular metabolites required to outcompete microbial foes in antagonistic interactions. Cutting-edge metabolomics analysis shed light on the bases of this antagonism, suggesting that mutations in ComQXPA provide a metabolic shift that resulted in two major outcomes: (i) post-ST variants were capable of retaining ability to produce effective levels of antifungal 2-ketones upon interaction with the fungus, while otherwise (ii) switching off the production of surfactin, which would otherwise interfere with the antifungal activity of 2-ketones. As a bonus, mutations in ComQXPA also promoted biofilm formation, another important feature to overcome biological interactions in natural habitats, such as the root of plants.

## Methods

**Microorganisms and culture conditions**. Bacterial and fungal strains used in this study are listed in Supplementary Table 2. Bacterial strains were routinely grown in LB medium (10 g/l tryptone, 5 g/l yeast extract, and 10 g/l NaCl) for 8 h at 30 °C with agitation. For coculture experiments and recovery of post-ST variants, a mycelial disc of 5 mm diameter taken from a 7-day-old culture of *S. terrestris* was placed in the center of a Petri dish containing 20 ml of PDA medium. After 4 days, an aliquot of a fresh culture of *B. subtilis* ALBA01 was inoculated at 2 cm distance on each side of the fungal colony. Petri dishes were incubated at 28 ± 2 °C. After 15 days of coculture, a sample of the bacterial colony was taken, stored, and named post-ST variant to differentiate it from its original version never exposed to the fungus in coculture (pre-ST variant). For biofilm assays, cells were grown in either LB or LB plus 1 % (v/v) glycerol and 100 μM MnSO₄ (LBGM) at 30 °C. When necessary, LB and LBGM media were solidified with 1.5% (w/v) agar. Detailed description of biofilm experimental procedures can be found in the Supplementary Methods section.

**Construction of mutant strains**. Integrative plasmid was constructed to disrupt *srfAA* gene. To construct pSG1194-srfAA, an internal fragment *srfAA* from *B. subtilis* ALBA01 were amplified using primer pairs (listed in Supplementary Table 3), containing EcoRI and BamHI restriction sites. PCR products were digested with EcoRI and BamHI, and ligated into pSG1194 vector digested with the same enzymes. Ligation mixtures were transformed into heat shock *Escherichia coli* DH5α competent cells. Plasmids were purified, and ALBA01 competent cells were transformed by standard protocols[38] to generate the mutant strain Bs *srfAA*. For confirmation of gene disruption and selection of single-copy transformants, chloramphenicol-resistant colonies were analyzed by PCR and subjected to Sanger sequencing (Unidad de Genómica, Instituto de Biotecnología, C.I.C.V.yA. Instituto Nacional de Tecnología Agropecuaria (INTA), Buenos Aires, Argentina).

**Whole-genome sequencing and analysis**. Whole-genome sequencing was performed using a paired-end (PE) 2× 100 bp library on Illumina Hiseq 1500 system (INDEAR Genome Sequencing facility, Argentina), and de novo assembly on A5

pipeline[39]. ALBA01 reads were assembled into 28 scaffolds with average scaffold size 147,128 bp. For annotation of the genome, scaffolds were uploaded to Rapid Annotation using Subsystem Technology (RAST) server[40], and SEED-based method was applied on this server. The resulting assembly was used as reference genome sequence to map reads from post-ST variants using BWA-MEM tool (v. 0.7.5a-r405)[41,42]. Variants were called using Genome Analysis Toolkit (GATK) HaplotypeCaller[43], and single-nucleotide polymorphisms (SNPs) were filtered based on Phred score >99%. SNPs detected from analysis of ancestral reads were excluded as false positives. Integrative Genomics Viewer (IGV)[44,45] was used for manual inspection of variants and read alignments. *comA* and *comP* gene mutations were validated to confirm their presence in post-ST variants and absence in pre-ST variants or ALBA01. For this, a 484-bp *comA* fragment was amplified by PCR using oligonucleotide primers FcomA_map and RcomA_map (Supplementary Table 3). To map mutations in *comP*, a PCR fragment was amplified using primer pair FcomP_map/RcomP_map. PCR products were purified with Silica Bead DNA Gel Extraction Kit (Thermo Fisher, USA), subjected to Sanger sequencing, and analyzed using BioEdit[46].

**Pooled whole-genome sequencing analysis**. Fifteen *B. subtilis* post-ST variants were obtained from independent coculture experiments, as described above. Genomic DNA was collected and purified from each clone using Promega Wizard Genomic DNA purification kit as per the manufacturer's protocol. DNA quantity was determined using Qubit Fluorometric Quantitation fluorometer (Thermo Fisher), and appropriate dilutions were mixed together such that each genome was represented equally in the final pool. Libraries were prepared using Nextera XT DNA Library Preparation Kit (Illumina, USA), as per the manufacturer's protocol. PE sequencing was performed on an Illumina MiSeq platform producing 2× 150 bp read lengths (INDEAR Genome Sequencing). Analysis of raw data quality was performed with FastQC[47]. Adapter sequences were trimmed using Trimmomatic[48]. Pool-seq reads were mapped against ALBA01 reference genome using BWA ALN[42]. PCR duplicates, multialignment reads, improperly paired aligned reads, and soft-clipped alignments were removed using Samtools[49]. Indels were realigned with GATK[43]. SNP, insertion, and deletion discovery was performed using HaplotypeCaller algorithm[43], with sample ploidy parameter set to 15. Under this methodology, variants detected by analyzing ALBA01 ancestral reads were excluded as potential false positives. Filter sequence variants was performed by running GATK VariantFiltration with parameters "QD < 2.0 || FS > 60.0 || MQ < 40.0 || MQRankSum < −12.5 || ReadPosRankSum < −8.0" for SNP, and "QD < 2.0 || FS > 200.0 || ReadPosRankSum < −20.0" for indels.

**HPLC–tandem mass spectrometry**. Biological triplicates of *B. subtilis* pre- and post-ST variants, as well as a *B. subtilis* NCBI 3610 wild-type strain and a *B. subtilis* NCBI 3610 knockout *srfAA* mutant were cultivated in LB medium at 30 °C at 180 r.p.m. The cultures were harvested after 18 h through centrifugation at 7000 r. p.m., 4 °C for 5 min. Cell metabolism was quenched by resuspending the cell pellet in −80 °C methanol to a final concentration of 500 mg/ml to final volume of 500 μl in a 96-deepwell plate. The samples were then sonicated 10 min and the cell debris was pelleted by 10 min centrifugation at 7000 r.p.m. A total of 450 μl of supernatants were transferred to a new deepwell plate and dried overnight in a CentriVap Benchtop Vacuum Centrifuge (LabConco). The metabolites were then resuspended in 100 μl 50% (v/v) methanol, 1% (v/v) formic acid (FA), and submitted for HPLC–MS/MS analysis. Nontargeted HPLC–MS/MS analysis was performed according to Petras et al.[50]. Therefore, 5 μl of the samples were injected on a Q-Exactive Quadrupole-Orbitrap mass spectrometer coupled to Vanquish ultrahigh-performance liquid chromatography system (Thermo Fisher Scientific, Bremen, Germany). For the LC separation, a C18 core-shell column (Kinetex, 50 × 2 mm, 1.8 μm particle size, 100 A pore size, Phenomenex, Torrance, USA) with a flow rate of 0.5 ml/min (solvent A: $H_2O$ + 0.1% (v/v) FA, solvent B: acetonitrile + 0.1% v/v/ FA) was used. During LC–MS/MS analysis, the compounds were eluted with a linear gradient from 0–0.5 min, 5% B, 0.5–4 min 5–50% B, 4–5 min 50–99% B, flowed by a 2 min washout phase at 99% B and a 2 min re-equilibration phase at 5% B. For positive mode MS/MS analysis, the electrospray ionization parameters were set to 35 l/min sheath gas flow, 10 l/min auxiliary gas flow, 2 l/min sweep gas flow, and 400 °C auxiliary gas temperature. The spray voltage was set to 3.5 kV and the inlet capillary was set to 250 °C. A S-lens voltage of 50 V was applied. MS/MS product ion spectra were recorded in data-dependent acquisition mode. Both MS1 survey scans (150–1500 *m/z*) and up to five MS/MS scans per duty cycle were measured with a resolution (*R*) of 17,500 with one micro-scan. The maximum C-trap fill time was set to 100 ms. Quadrupole precursor selection windows were set to 1 *m/z*. Normalized collision energy was stepwise increased from 20 to 30 to 40% with $z = 1$, as default charge state. MS/MS scans were automatically triggered at the apex of chromatographic peaks within 2–15 s from their first occurrence. Dynamic exclusion was set to 5 s. Ion species with unassigned charge states and isotope peaks were excluded from MS/MS acquisition.

Detailed descriptions of data analysis, MS/MS network analysis, and metabolomics studies based on NMR spectroscopy are presented in the Supplementary Information section.

**Gas chromatography–mass spectrometry metabolomics.** Biological replicates of *B. subtilis* pre- and post-ST variants and of the fungus *S. terrestris* were cultivated individually and in cocultures in 10-ml borosilicate vials with a screw cap with silicon septum containing 3 ml of PDA medium for 7 days at 30 °C, and then analyzed with GC–MS to determine the emitted volatile metabolites. After incubation, the capped vials were stored at −80 °C and thawed immediately prior to analysis. The GC–MS analysis was carried out using the Agilent 7200 GC/QTOF equipped with robotic sampler system. The volatiles from the sample were extracted from headspace using polydimethylsiloxane/divinylbenzene df 65 μm SPME fiber for 30 min at 50 °C. The fiber was then inserted into the injector equipped with Merlin septum heated to 250 °C and the adsorbed compounds were desorbed for 1 min. The GC protocol analysis included: cryofocusing on the head of the column at −20 °C for 1 min; 115 °C/min oven ramp to 40 °C (hold of 0.1 min), 20 °C/min oven ramp to 300 °C (hold of 0.1 min), ramp to 320 °C, and 50 °C/min oven ramp to 320 °C purge the column. The helium carrier gas was set to constant 2 ml/min flow and a splitless injection mode was applied. The scanned *m/z* range was 35–400 Th with the acquisition rate of 20 spectra/s. The empty vial blanks were interspersed with the samples to assess background signal. Quality controls of natural mint oil extract were run along with samples before and after the analysis. The GC–MS data were processed with MZmine2 (https://bmcbioinformatics.biomedcentral.com/articles/10.1186/1471-2105-11-395), using ADAP algorithm (https://pubs.acs.org/doi/full/10.1021/acs.jproteome.7b00633?src=recsys) deployed on the ProteoSAFE workflow of the GNPS platform (gnps.ucsd.edu). The parameters were set as indicated in the Supplementary Table 4. Data were uploaded to GNPS and searched against NIST 2017 and WILEY spectral libraries (link to GNPS search results: https://gnps.ucsd.edu/ProteoSAFe/status.jsp?task=9569f1598a414e9a926721942e41ed6c). The GC–MS data are available at the MassIVE depository (massive.ucsd.edu) under ID MSV000083294.

**Statistics and reproducibility.** For HPLC–MS/MS metabolomics data, tridimensional PCoA plots of MS1 data were generated by in-house tool ClusterApp using Bray–Curtis distance metric. Resulting scatter plots were visualized on EMPeror. Random forest classification was performed in R. For GC–MS 2-ketones data, boxplots were generated with abundances normalized by quantile normalization and auto-scaled (zero-centered and divided by SD). In suppressor effect of exogenous surfactin on the antifungal activity of cell-free supernatants of post-ST and of 2-ketones assays, one-way analysis of variance were performed followed by Tukey's multiple comparison test. For quantification of the biofilm formation and antifungal activities of cell-free supernatants of Bs ALBA01 pre-ST, post-ST, and Bs *srfAA* mutant, significant difference between values was evaluated with Tukey's multiple comparison test. Statistically significant differences at $p < 0.0001$, $p < 0.001$, and $p < 0.05$ were identified by ***, **, and *, respectively. One-way analysis of variance and Tukey's multiple comparison test were done in GraphPad Prism version 8.0.0 software. For $^1$H-NMR, spectral data were imported to SIMCA (v. 14.1, Umetrics AB; Umeå, Sweden) for multivariate data analysis: principal component analysis, using the Pareto-scaled NMR dataset, and OPLS-DA.

**Reporting summary.** Further information on research design is available in the Nature Research Reporting Summary linked to this article.

## Data availability

MS/MS data availability: all HPLC–MS/MS data can be found on the Mass spectrometry Interactive Virtual Environment (MassIVE) at https://massive.ucsd.edu/ with the identifier MSV000082081. The GC–MS data are available at the MassIVE depository (massive.ucsd.edu) under ID MSV000083294. Nucleotide sequence accession numbers: sequence of ALBA01 assembled genome was deposited in NCBI database (Bioproject PRJNA316980)[26]. Genome sequence reads from post-ST variants and Pool-seq analysis were deposited in NCBI database (Bioproject PRJNA480851). All relevant data are available from A.G.A.O. and A.M.S. upon request. Source data underlying graphs and charts presented in Figs. 1–4, and Supplementary Figs. 2, 4 and 5, is available in Supplementary Data 1 and Supplementary Data 2, respectively.

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

## Acknowledgements

This paper is dedicated to the memory of our colleague Dr. Daniel A. Ducasse, who passed away in May 2020. Dr. Ducasse was co-author of the present work and director of A.G.A.O. scientific career. This study was supported by MINCyT-Agencia Nacional de Promoción Científica y Tecnológica (ANPCyT; grants PICT 2013-2592 to A.G.A.O and PICT 2016-1545 to A.M.S.), U.S. National Science Foundation (NSF) Inspire Track II (grant IOS-1343020), National Institute of Health (NIH; grants GMS10RR029121 and 5P41GM103484–07), and Deutsche Forschungsgemeinschaft (DFG; grant PE 2600/1). The authors are grateful to Dr. Zdenek Kamenik, Institute of Microbiology of the Czech Academy of Sciences, for kindly providing ketones compounds, to Consejo Nacional de Investigaciones Científicas y Técnicas (CONICET) and Fulbright Commission Argentina.

## Author contributions

A.G.A.O. and A.M.S. designed the experiments and supervised the study. A.G.A.O, F.J, and P.S. cultured the strains. A.G.A.O. performed interaction experiments. A.G.A.O., R.A.T., A.J.M., and A.M.S. performed DNA isolation and analyzed bioinformatic data of bacterial genomes. A.G.A.O. analyzed NMR data. A.G.A.O., D.P., and A.A.A. collected MS data. A.G.A.O., D.P., A.A.A., M.W., and P.C.D. analyzed metabolomics data and performed molecular networking. A.G.A.O. and A.M.S. integrated all experimental data and made final interpretations. A.G.A.O., A.J.M., A.M.S. wrote the manuscript. All authors discussed, edited, and approved the finalized manuscript.

## Competing interests

P.C.D. is a scientific advisor for Sirenas LLC. M.W. is a consultant for Sirenas LLC and the founder of Ometa labs LLC. A.A.A. is a consultant for Ometa labs LLC. A.G.A.O., D.P., R.A.T., P.S., A.J.M., and A.M.S. have no competing interests to declare.
