## [Peer Review File · Communications Biology]

Reviewers' comments:

Reviewer #1 (Remarks to the Author):

This manuscript describes how quorum sensing system adapts during fungal-bacterial interaction. Interaction of *B. subtilis* to a soil fungal pathogen, *S. terrestris* results in mutations in the ComQ_{PA} quorum-sensing system which leads to changes in morphology, increases in biofilm formation, loss of swarming and hemolytic activity, decreased lipopeptides surfactin and plipastin, and more importantly increased production of 2-ketones with anti-fungal activity.

The contribution of this manuscript in understanding the effect of fungal-bacterial interaction on quorum sensing systems and bacterial metabolism is significant. Enough data were provided to support the conclusion. The manuscript was clear in communicating the main findings of the study, but some parts were confusing.

Addressing the following issues could help in clarifying those confusions:

Major comments:

1. Surfactin coded by *srfAA* interferes with the anti-fungal activity of 2-ketones. The data seem to suggest surfactin decreases production or stability of ketones. Was 2-ketone production measured in a *srfAA* mutant? If these data are available, please include them and indicate how ketone levels compare to that of a pre-ST and a post-ST variant? Lines 174 – 176 state that this interference has been reported previously – was the mechanism of interference previously explored?
2. In lines 63 – 64, the authors mentioned that surfactin is important for biofilm formation, however post-ST variants have enhanced biofilm formation (Supplementary Fig. 1c) which is also claimed to be a phenotype associated with ComQ_{PA} mutant (lines 232 – 234). These contradicting results were also mentioned in lines 127 – 129, but not explained. It would be great to clarify this or include an explanation of the contradicting statements/results.
3. Fig. 2b shows that 2-ketone abundance dropped sharply when *S. terrestris* is cocultured with pre-ST but not with post-ST (lines 147 – 148). Is there a potential explanation for this observation?

Minor comments:

1. Line 37 – typo error: must be spelled *Bacillus*
2. Line 61 – typo error: must be spelled lipopeptides
3. Line 81 – Supplementary Fig. 1a must be Supplementary Fig. 1b-c
4. Line 144 – “were the also among top” must be “were also among the top”
5. Line 145 – typo error: must be Fig. 2b
6. Was *srfAA* disrupted or deleted? Construction of mutant strains section suggests that the *srfAA* gene was disrupted and not deleted, while line 119 states that the gene was deleted. This should be made clear in many places throughout the manuscript (including Supplementary Fig. 6 and Table 2) whether the strain used was a *srfAA::xx* or a Δ *srfAA*.
7. Line 169: must be Fig. 3a
8. Line 308 – Is *B. subtilis* NCBI 3610 the same as *B. subtilis* ALBA01? This should be made clear.
9. Line 358: “and a),” is either a typo error or something is missing in that sentence
10. Fig. 1 – Typo error in Fig 1a: “Srf-defficient” should be Srf-deficient. For Fig. 1C and D, a better legend to differentiate between post-ST1 and post-ST2 would help. I don't see these 2 in both graphs. Are they below the detection level? Maybe, include a note about that in the legend. Also, what are post-ST1, 2, and 3? A description in the legend would also help especially that these 3 were described not until at the latter part of the paper.
11. Line 535 – typo error: duplicated “in”

12. Line 544 – delete comma after both
13. Fig 3 – add details on the statistical analysis in the caption
14. Supplementary Fig. 1 – For clarity, the plates in Fig 1d could be labeled with pre- and post-ST as well. Fig. 1d was not also mentioned in the text.
15. Supplementary Fig. 2b – Identifying which corresponds to pre-ST and post-ST in this figure could help in the understanding of those who are not familiar with this technique.
16. Supplementary Fig. 4a and b – same concern on legend used for post-ST1 and post-ST2 in Fig. 1
17. What is the difference between Fig.1 C, D and Supplementary Fig. 4A, B? Would it be possible to just show either one of them for simplicity?
18. Supplementary Fig. 7 – The details on the stat analysis should be included in the caption.
19. Line 794 – variants 3 and 4 should be variants 2 and 3
20. Supplementary Table 2 – the table border must be modified
21. Supplementary Fig. 1, 4d, and 8 are all showing phenotypic characterization. Could all these be combined?
22. Supplementary Fig. 2 and 3 are also related. Could these two be combined as well?
23. In many places throughout the manuscript, be consistent with the abbreviations for measurements (e.g. mL vs. ml)
24. In many of the figures, it would be great to describe number of replicates or if representative data is being shown.

Reviewer #2 (Remarks to the Author):

In previous work, after extended co-culture with the phytopathogenic fungi *S. terrestris*, the authors isolated several *B. subtilis* strains with increased antifungal activity. In this work, using a combination of metabolomics, genomics and classic genetics the authors identified the depletion of Surfactin as the basis for the increase in antifungal activity. Furthermore, they were able to demonstrate that this is due to an inhibitory effect of surfactin on the activities of 2-heptanone and 2-octanone. Although complementation of a mutant is standard practice to assign a phenotype, I consider that the independent obtention of mutants of genes related to regulation/synthesis of Surfactin, with similar phenotypes, provides robust evidence to support the conclusions.

1) My major issue with this manuscript is its lack of clarity derived from its structure. After the Abstract there are no more sections named: no Introduction, Results or Discussion sections, everything is fused in one title-less text wall. More structure in the text is needed to increase clarity. It is ok to fuse the results and discussion sections but subtopic, clearly defined by titles and subtitles, that allow to better follow the narrative of the manuscript should be added. This will also help to clearly differentiate the work reported in reference 15 from the work performed for this manuscript.

2) A deeper discussion of the biological implications of the conclusions may improve this work. Your results show that individually cultured pre-ST variants showed high levels of 2-ketones. However, co-culture with *S. terrestris* was able to induce a steep drop in 2-ketones. Did you consider the possibility that this can be explained by chemical interference from a fungal secreted compound? This is especially interesting because from the several strains that were sequenced you identified several mutations in different members from TCSTS. Alternatively, the induction of surfactin may be a response to achieve the rapid colonization of solid media through swarming. It may be interesting to measure 2-ketones levels under biofilm-forming conditions.

Minor Issues

There can be confusion regarding figures 1C and 1D and supplementary figures 4A and 4B. Both images have the same name and descriptions are also the same, but one figure shows results from whole cell and the other from cell-free supernatants. Make this clear in supplementary figure 4 legend.

Reviewer #3 (Remarks to the Author):

This manuscript investigates the impact of incubating the soil bacterium *B. subtilis* with a fungal isolate called *S. terrestris*. Heritable alterations to the *B. subtilis* genome are identified after co-incubation and the impact on the metabolome are defined using next generation sequencing. The mutations acquired in the ComQXPA system alter *B. subtilis* metabolism such that surfactin production declines and production of antifungal volatiles increases. The manuscript is interesting and is well presented. The work is novel and highlights the impact that mixing microbial species can have at the molecular level. It is of interest to the field.

Points for attention that most likely arise from the brevity of the manuscript are detailed below:

It would be helpful to show the individual points on bar charts such as that in Fig. S6 and 7 (and other charts throughout). This increases the transparency of the work.

Please include more details about how the two isolates were incubated. This seemed to be lacking beyond that they were incubated for 15-days. Was this in liquid media, on a plant, on a plate, etc.?

Ensure all methods that were used are detailed- eg SEM analysis method seems to be lacking. What growth media were the fungal isolates grown on?

Check that all strains are detailed in the strain table, NCIB3610 and srfAA derivative are not detailed (although noted in the methods).

Ensure all figures and subparts are cross referenced in the text and that the figure legends are full and completed. For example, Fig. 3 and Fig. S7 do not detail the plates that are shown or detail what the abbreviations define; Fig. S4 d- needs more detail of the images – scale, incubation time, agar etc.

Line 81- has the incorrect figure reference.

Add w/v or v/v as needed in the manuscript when referring to percentages.

COMMSBIO-20-0821 "Fungal-bacterial interaction selects for quorum sensing mutants and a metabolic shift towards the production of natural antifungal compounds" Andrea G. Albarracín Orio *et al*

Response to Reviewers

Reviewer #1:

This manuscript describes how quorum sensing system adapts during fungal-bacterial interaction. Interaction of *B. subtilis* to a soil fungal pathogen, *S. terrestris* results in mutations in the ComQPXA quorum-sensing system which leads to changes in morphology, increases in biofilm formation, loss of swarming and hemolytic activity, decreased lipopeptides surfactin and plipastin, and more importantly increased production of 2-ketones with anti-fungal activity.

The contribution of this manuscript in understanding the effect of fungal-bacterial interaction on quorum sensing systems and bacterial metabolism is significant. Enough data were provided to support the conclusion. The manuscript was clear in communicating the main findings of the study, but some parts were confusing.

Response:

We appreciate the Reviewer's comments on the work. We have gone through all the Reviewer points and suggestions and incorporated the changes requested.

Addressing the following issues could help in clarifying those confusions:

Major comments:

1. Surfactin coded by *srfAA* interferes with the anti-fungal activity of 2-ketones. The data seem to suggest surfactin decreases production or stability of ketones. Was 2-ketone production measured in a *srfAA* mutant? If these data are available, please include them and indicate how ketone levels compare to that of a pre-ST and a post-ST variant? Lines 174 – 176 state that this interference has been reported previously – was the mechanism of interference previously explored?

Response:

The GC-MS and molecular networking experiments, which allowed us to measure 2-ketones were carried out for the *B. subtilis* ALBA01 pre and post-ST variants as well as their respective co-cultures with the fungus. We have not measured production of 2-ketones in the *srfAA* mutant but we fully agree that this will be *a must* in our future post-quarantine plans to keep disentangling the molecular mechanisms and physicochemical bases of surfactin interference.

To date, a couple of reports have only provided hypothetical explanations on how surfactin may interfere with antifungal lipopeptidic compounds such as fengycins and iturins. Yet, the underlying mechanisms of surfactin interference with 2-ketones or other compounds have not been explored.

Kim *et al.* (reference 25) reported that a surfactin-deficient mutant of *B. subtilis* subsp. *krietiensis* ATCC55079 showed significantly higher antifungal activity against the fungus *Fusarium oxysporum* than its parental wild-type strain. This surfactin-deficient mutant, carrying a disrupted surfactin biosynthesis gene, exhibited a 30% increase in the levels of lipopeptide iturin A compared to the wild-type strain. The authors suggested that higher antifungal activity was achieved due to enhanced production of iturin A and, that surfactin could be either blocking iturin production by sequestering substrates and/or by changing the activity of positive/negative regulators of cyclic lipopeptides.

The work of Zihahirwa Kulimushiet *al.* (reference 13) illustrates the efficacy of *Bacillus amyloliquefaciens* strains at inhibiting *Rhizomucor variabilis*, a fungal pathogen of maize plants. Based on the loss of function of specifically repressed mutants and on the activity of purified compounds, they showed that fengycin-type lipopeptides provide the main antifungal potential against *R. variabilis*. Mutants overproducing fengycins and/or iturins but not surfactin displayed higher antifungal activity. By testing mutants selectively repressed in the synthesis of one or two lipopeptides, they revealed that the co-production of surfactin together with fengycin decreased the global antifungal activity. They speculate that fengycins may be involved in permeabilization of spore/conidia and, therefore, inhibiting germination and/or causing hyphal cell perturbation. Opposing to this, surfactin may have a stabilizing effect on certain lipid bilayers, thus limiting pore formation in fungal membranes. They also propose that under certain conditions and concentrations, surfactins and fengycins may co-aggregate and form inactive complexes, thereby decreasing antifungal activity.

In the revised version of the manuscript, we include a more detailed discussion of both references in relation to our results (Pages 9 and 10, lines 228 to 254).

2. In lines 63 – 64, the authors mentioned that surfactin is important for biofilm formation, however post-ST variants have enhanced biofilm formation (Supplementary Fig. 1c) which is also claimed to be a phenotype associated with ComQXPA mutant (lines 232 – 234). These contradicting results were also mentioned in lines 127 – 129, but not explained. It would be great to clarify this or include an explanation of the contradicting statements/results.

Response:

We agree with reviewer comment that this observation may, at first, result intriguing and contradicting. The ComQXPA system is involved in the expression of surfactin-related genes, and surfactin has been described as a signal molecule involved in the activation of biofilm pathways. Therefore, it is expected that the inactivation of the ComQXPA system leads to decreased surfactin production. It is widely stated/accepted that surfactin plays a crucial role in biofilm formation. However, it has been recently described that *B. subtilis* mutants in genes of the ComQXPA quorum-sensing system (QS) can actually form pellicles with more matrix components than the wild type strain (Spacapan *et al.*, 2019, reference 36). These mutant strains are able to develop earlier and thicker biofilms than the wild type strain, even though with decreased surfactin production. In agreement with this, we suggest that post-ST variants carrying loss of function mutations in ComQXPA system, which result in inactivation of QS, impact in a similar fashion as described by Spacapan *et al.*, promoting biofilm formation as one of their adaptive strategies to overcome interaction with the fungus. In the revised version of

the manuscript, we included this new reference and a brief discussion (Pages 12 and 13, lines 308 to 330).

On the other hand, Oslizlo and co-workers (2015, *Microbial Biotechnology*, reference 37) explored the ComQXPA diversity among isolates from the tomato rhizoplane, another natural habitat of *B. subtilis*. They found that rhizoplane isolates showed a remarkable diversity in surfactin production and that this production was not always positively correlated to biofilm formation. They observed that *urfA* gene inactivation in five isolated strains showed different effects on biofilm biomass production: two mutants displayed decreased levels of biofilm formation (28-32% reduction), whereas two mutants formed biofilm biomass comparable to their wild type ancestor and, one mutant showed a 3.5 fold increased biomass respect to the parental strain. Therefore, they conclude that the regulatory role of surfactin in biofilm formation may vary among different natural isolates of *B. subtilis* such as our *B. subtilis* ALBA01 strain. In the revised version of the manuscript, we included this reference and a brief discussion (Page 12, lines 316 to 330).

3. Fig. 2b shows that 2-ketone abundance dropped sharply when *S. terrestris* is co-cultured with pre-ST but not with post-ST (lines 147 – 148). Is there a potential explanation for this observation?

Response:

We found this result intriguing and we believe that, because of the physical-chemical properties of surfactin molecule, it may be interfering with 2-ketone activity by a direct chemical reaction altering ketone's molecule and/or, by physical sequestration through micelle formation. We also speculate that *S. terrestris* may be using the surfactant milieu provided by surfactin to deliver chemical signals to induce a reduction of 2-ketone production by *B. subtilis*. This possibility is in agreement with the fact that surfactin-producer pre-ST variants show the highest production of 2-ketones when they are not interacting with *S. terrestris*. Thus, post-ST variants, which do not produce surfactin, would be less susceptible of being targeted by these fungal signals with the ability to shut down production of 2-ketones. These hypotheses still need to be tested and we are planning to do so in the close future.

Minor comments:

1. Line 37 – typo error: must be spelled Bacillus

Response:

The error has been corrected in the revised version of the manuscript.

2. Line 61 – typo error: must be spelled lipopeptides

Response:

The error has been corrected in the revised version of the manuscript.

3. Line 81 – Supplementary Fig. 1a must be Supplementary Fig. 1b-c

Response:

The error has been corrected in the revised version of the manuscript.

4. Line 144 – “were the also among top” must be “were also among the top”

Response:

The error has been corrected in the revised version of the manuscript.

5. Line 145 – typo error: must be Fig. 2b

Response:

The error has been corrected in the revised version of the manuscript.

6. Was *srfAA* disrupted or deleted? Construction of mutant strains section suggests that the *srfAA* gene was disrupted and not deleted, while line 119 states that the gene was deleted. This should be made clear in many places throughout the manuscript (including Supplementary Fig. 6 and Table 2) whether the strain used was a *srfAA::xx* or a Δ *srfAA*.

Response:

The *B. subtilis* ALBA01 surfactin-deficient mutant was obtained by disruption of the *srfAA* gene. We corrected this error throughout the revised version of manuscript in order to clarify this point.

7. Line 169: must be Fig. 3a

Response:

The error has been corrected in the revised version of the manuscript.

8. Line 308 – Is *B. subtilis* NCBI 3610 the same as *B. subtilis* ALBA01? This should be made clear.

Response:

The genetic background of *B. subtilis* NCBI 3610 surfactin-deficient mutant is the *B. subtilis* reference laboratory strain NCBI 316 and therefore, is different from *B. subtilis* ALBA01 *srfAA* (named *Bs srfAA* mutant in the revised version of the manuscript), whose genetic background corresponds to the *B. subtilis* ALBA01 strain characterized in our laboratory. We have clarified this point in the revised manuscript, and the NCBI 3610 surfactin-deficient mutant and its source was included in Supplementary Table 2 and named *B. subtilis* NCBI 3610 *srf*-deficient strain.

9. Line 358: “and a),” is either a typo error or something is missing in that sentence

Response:

The error has been corrected in the revised version of the manuscript.

10. Fig. 1 – Typo error in Fig 1a: “Srf-defficient” should be Srf-deficient.

Response:

The error has been corrected in the revised version of the manuscript.

For Fig. 1C and D, a better legend to differentiate between post-ST1 and post-ST2 would help. I don't see these 2 in both graphs. Are they below the detection level? Maybe, include a note about that in the legend. Also, what are post-ST1, 2 and 3? A description in the legend would also help especially that these 3 were described not until at the latter part of the paper.

Response:

We thank the Reviewer for this observation. As requested, we have included a detailed description of the co-culture process with the fungus and of the methodological steps employed to obtain post-ST variants in the Material and Method section (Pages 13 and 14, lines 357 to 364). We have also included a phrase in Figure 1 legend explaining the origin of the different post-ST variants and their metabolomics profiles compared to the pre-ST variant.

11. Line 535 – typo error: duplicated “in”

Response:

The error has been corrected in the revised version of the manuscript.

12. Line 544 – delete comma after both

Response:

The error has been corrected in the revised version of the manuscript.

13. Fig 3 – add details on the statistical analysis in the caption

Response:

As requested, we added details on the statistical analysis for the experiment showed in Figure 3.

14. Supplementary Fig. 1 – For clarity, the plates in Fig 1d could be labeled with pre- and post-ST as well. Fig. 1d was not also mentioned in the text.

Response:

We have added labels to Figure 1S according to the Reviewer's suggestion. Furthermore, we have also added the respective cross-reference for the Figure 1d, which was lacking in the text.

15. Supplementary Fig. 2b – Identifying which corresponds to pre-ST and post-ST in this figure could help in the understanding of those who are not familiar with this technique.

Response:

To make OPLS DA S line plots more understandable, we included more detailed aspects about their interpretation in the Figure 2b legend of Supplementary Material.

16. Supplementary Fig. 4a and b – same concern on legend used for post-ST1 and post-ST2 in Fig. 1

Response:

To clarify this point, a phrase was included in the Figure legend, which in the revised version of the manuscript is Supplementary Figure 3a and b.

17. What is the difference between Fig.1 C, D and Supplementary Fig. 4A, B? Would it be possible to just show either one of them for simplicity?

Response:

We thank the Reviewer for this observation. Figure 1c-d shows the intensities of typical fragmentation ions of lipopeptides surfactin (c) and plipastatin (d) as evaluated by LC-MS/MS. This experiment was performed using cell-free supernatants of pre-ST and post-ST variants 1, 2 and 3. On the other hand, Supplementary Figure 4 a-b shows the intensities of surfactin and plipastatin ions from whole cells samples (pellets cells from cultures) of each variant. Results from both approaches are consistent and show the inability of post-ST variants to produce and release surfactin and plipastatin. So, we maintained both Figures in the revised version and added a sentence in the legend of this Supplementary Figure, which is now Supplementary Figure 3, to clarify this point and differentiate it from the results shown in Figure 1.

18. Supplementary Fig. 7 – The details on the stat analysis should be included in the caption.

Response:

Following the reviewer suggestion, details on the statistical analysis for experiments shown in the Figure 7 were included.

19. Line 794 – variants 3 and 4 should be variants 2 and 3

Response:

As pointed by the Reviewer, the error has been corrected in the revised version of the manuscript.

20. Supplementary Table 2 – the table border must be modified

Response:

As requested, the table border in Supplementary Table 2 has been modified in the revised version of the manuscript.

21. Supplementary Fig. 1, 4d, and 8 are all showing phenotypic characterization. Could all these be combined?

Response:

As suggested, Supplementary Figures 1, 4d and 8 have been combined in the revised version of the manuscript.

22. Supplementary Fig. 2 and 3 are also related. Could these two be combined as well?

Response:

As suggested, Supplementary Figures 2 and 3 have been combined in the revised version of the manuscript.

23. In many places throughout the manuscript, be consistent with the abbreviations for measurements (e.g. mL vs. ml)

Response:

As suggested, we went through the manuscript and corrected errors and inconsistencies related to units and abbreviations.

24. In many of the figures, it would be great to describe number of replicates or if representative data is being shown.

Response:

As suggested, we included the data about the number of replicates used for each of the experiments shown in figures.

Reviewer #2:

In previous work, after extended co-culture with the phytopathogenic fungi *S. terrestris*, the authors isolated several *B. subtilis* strains with increased antifungal

activity. In this work, using a combination of metabolomics, genomics and classic genetics the authors identified the depletion of Surfactin as the basis for the increase in antifungal activity. Furthermore, they were able to demonstrate that this is due to an inhibitory effect of surfactin on the activities of 2-heptanone and 2-octanone. Although complementation of a mutant is standard practice to assign a phenotype, I consider that the independent obtention of mutants of genes related to regulation/synthesis of Surfactin, with similar phenotypes, provides robust evidence to support the conclusions.

Response:

We appreciate the Reviewer's comments on the work. We have gone through all the Reviewer points and suggestions and incorporated the changes requested.

1) My major issue with this manuscript is its lack of clarity derived from its structure. After the Abstract there are no more sections named: no Introduction, Results or Discussion sections, everything is fused in one title-less text wall. More structure in the text is needed to increase clarity. It is ok to fuse the results and discussion sections but subtopic, clearly defined by titles and subtitles, that allow to better follow the narrative of the manuscript should be added. This will also help to clearly differentiate the work reported in reference 15 from the work performed for this manuscript.

Response:

To clarify and clearly differentiate the results shown in the present work respect to those from reference 17, the revised version of the manuscript was re-structured by adding a new separate and extended Introduction section, and subheadings were introduced in the combined Results and Discussion section. We have also incorporate a *Concluding remarks* paragraph to highlight the main results and message of this work (Page 13, lines 332 to 347).

2) A deeper discussion of the biological implications of the conclusions may improve this work. Your results show that individually cultured pre-ST variants showed high levels of 2-ketones. However, co-culture with *S. terrestris* was able to induce a steep drop in 2-ketones. Did you consider the possibility that this can be explained by chemical interference from a fungal secreted compound? This is especially interesting because from the several strains that were sequenced you identified several mutations in different members from TCSTS. Alternatively, the induction of surfactin may be a response to achieve the rapid colonization of solid media through swarming. It may be interesting to measure 2-ketones levels under biofilm-forming conditions.

Response:

We agree with the Reviewer that it would be really interesting to address if there is a fungal interference through secretion of metabolites able to decrease 2-ketones levels, which could help to explain why pre-ST variants were unable to counteract and maintain effective/active concentrations/levels of these ketones. We find this aspect intriguing. It is possible to speculate that *S. terrestris* may be using the surfactant milieu provided by surfactin to deliver chemical signals to induce a reduction of 2-ketone production by *B. subtilis*. This possibility is in agreement with the fact that surfactin-producer pre-ST variants show the highest

production of 2-ketones when they are not interacting with *S. terrestris*. Thus, post-ST variants, which do not produce surfactin, would be less susceptible of being targeted by these fungal signals with the ability to interfere production of 2-ketones. Nevertheless, our evidences also suggest that surfactin was directly involved as well. Since post-ST variants (variants carrying mutations in ComQXPA system, involved in surfactin biosynthesis) can maintain effective levels of 2-ketones when interacting with the fungus, we conclude that surfactin was interfering with 2-ketone activity, probably by direct chemical reaction and/or by physical sequestration through micelle formation. Importantly, these results were supported by the experiments with combined purified surfactin and 2-ketones. Alternatively, we hypothesize that surfactin may have a stabilizing effect on the lipid bilayers in the fungal membrane generating some kind of resistance to the action of ketones. Nevertheless, we believe that a more complex or a combination of processes can be occurring. Respect to the Reviewer's comment, this manuscript is focused on a *B. subtilis* point of view, but future work involves deeper analyses on fungal metabolites to elucidate if any of these are able to trigger the *B. subtilis* response.

Then, as suggested by the Reviewer, a paragraph providing additional alternative biological implications of such observations was included in the revised version of the manuscript (Pages 9 and 10, lines 228 to 254).

Minor Issues

There can be confusion regarding figures 1C and 1D and supplementary figures 4A and 4B. Both images have the same name and descriptions are also the same, but one figure shows results from whole cell and the other from cell-free supernatants. Make this clear in supplementary figure 4 legend.

Response:

We thank the Reviewer for this observation. This point was also addressed by Reviewer 1. To clarify, we specified in figure titles as well as in their respective legends that Figures 1C and 1D correspond to cell-free supernatants of pre-ST and post-ST variants 1, 2 and 3, whereas Figures 4A and 4B display results from whole cells samples (pellets cells from cultures).

Reviewer #3:

This manuscript investigates the impact of incubating the soil bacterium *B. subtilis* with a fungal isolate called *S. terrestris*. Heritable alterations to the *B. subtilis* genome are identified after co-incubation and the impact on the metabolome are defined using next generation sequencing. The mutations acquired in the ComQXPA system alter *B. subtilis* metabolism such that surfactin production declines and production of antifungal volatiles increases. The manuscript is interesting and is well presented. The work is novel and highlights the impact that mixing microbial species can have at the molecular level. It is of interest to the field.

Response:

We appreciate and thank Reviewer's comments on the work. We went through all Reviewer points and suggestions and incorporated the requested changes.

Points for attention that most likely arise from the brevity of the manuscript are detailed below:

It would be helpful to show the individual points on bar charts such as that in Fig. S6 and 7 (and other charts throughout). This increases the transparency of the work.

Response:

Following the Reviewer's suggestion, details on the statistical analysis for experiments showed in Supplementary Figures 6 and 7 were added.

Please include more details about how the two isolates were incubated. This seemed to be lacking beyond that they were incubated for 15-days. Was this in liquid media, on a plant, on a plate, etc.?

Response:

As requested, details on the different microbiological procedures were included in the Material and Methods section of the revised manuscript (Pages 13 and 14, lines 351 to 363).

Ensure all methods that were used are details- eg SEM analysis method seems to be lacking. What growth media were the fungal isolates grown on?

Response:

As requested, detailed information about the Electron Microscopy analysis in the Material and Methods of the Supplementary Material section (Pages 2 and 3 , lines 53 to 61) was added in the revised version of the manuscript.

Check that all strains are detailed in the strain table, NCIB3610 and srfAA derivative are not detailed (although noted in the methods).

Response:

As requested, the NCBI 3610 surfactin-deficient mutant (named as *B. subtilis* NCBI 3610 *srf-deficient* strain) and its source were included in Supplementary Table 2 in the revised manuscript.

Ensure all figures and subparts are cross referenced in the text and that the figure legends are full and completed. For example, Fig. 3 and Fig. S7 do not detail the plates that are shown or detail what the abbreviations define; Fig. S4 d- needs more detail of the images – scale, incubation time, agar etc.

Response:

We thank the Reviewer for this observation. As suggested, cross-references for figures and their respective subparts were checked and added in the revised version. Also, details on labels and scales were included throughout the main text and figures.

Line 81- has the incorrect figure reference.

Response:

The error has been corrected in the revised version of the manuscript.

Add w/v or v/v as needed in the manuscript when referring to percentages.

Response:

As requested, units w/v or v/v have been added in the revised version of the manuscript.

Reviewers' comments:

Reviewer #1 (Remarks to the Author):

The reviewer adequately addressed all of the criticisms of the prior version of this manuscript.

There were only a few very minor corrections (typos) to note for this version:

line 112: delete comma

line 758-759: spacing

Supplementary line 50: 0.2 u?

Reviewer #2 (Remarks to the Author):

I consider that the comments and observations from the reviewers have been successfully addressed in the revised manuscript

Reviewer #3 (Remarks to the Author):

The authors have improved the manuscript and I am happy with the majority of the changes. The methods section and legends still need more work to make them fully comprehensive. The microbiological aspects of the work are perhaps skimmed over in certain ways. I could not find details of the agar plates or method used for the hemolysis assay, for example. A careful check to ensure all methods are documented is needed.

COMMSBIO-20-0821 B "Fungal-bacterial interaction selects for quorum sensing mutants and metabolic shift towards the production of natural antifungal compounds" Andrea G. Albarracín Orio *et al*

Response to Reviewers

Reviewer #1:

The reviewer adequately addressed all of the criticisms of the prior version of this manuscript.

There were only a few very minor corrections (typos) to note for this version:

Line 112: delete comma

Line 758-759: spacing

Supplementary line 50: 0.2 u?

Response:

We appreciate the Reviewer's comments on the revised version of the manuscript. We have gone through all the Reviewer points and incorporated the changes requested.

Reviewer #2:

I consider that the comments and observations from the reviewers have been successfully addressed in the revised manuscript

Response:

We appreciate the Reviewer's comments on the revised version of the manuscript.

Reviewer #3:

The authors have improved the manuscript and I am happy with the majority of the changes. The methods section and legends still need more work to make them fully comprehensive. The microbiological aspects of the work are perhaps skimmed over in certain ways. I could not find details of the agar plates or method used for the hemolysis assay, for example. A careful check to ensure all methods are documented is needed.

Response:

We appreciate and thank Reviewer's comments on the work. We went through the manuscript and incorporated extended information on microbiological methods as requested. Additional details on different microbiological procedures such as hemolytic activity, biofilm formation quantification, general media used and bacterial grow conditions, were included in the Material and Methods section of the revised manuscript (Page 13, line 351; Page 14, line 359-360) and revised Supplementary Methods (Page 2, lines 37 to 56).